

# Assessment of avalanche hazard situation in Turkey during years 2010s

## Tayfun Kurt

Independent researcher, Istanbul, Turkey, tyfnkrt@hotmail.com

**ABSTRACT**

Avalanches constitute risky situations especially for mountainous areas in the eastern part of Turkey. According to records of the Disaster and Emergency Management Presidency, avalanches have killed 30 people per year in Turkey over the last 30 years. Developing winter tourism also affects losses. For example, an avalanche occurred in *Torul, Köstere,* in the province of Giresun, on January 25, 2009, which killed 10 mountaineers and injured 7 people.

This research is focused on, known fatal avalanches and avalanche mitigation works. The obtanied map provides are reliable and easy to understand information where avalanches contstitue risksy sitation in regional scale as well as where new avalanche paths may develop under favourable conditions. Moreover the figure of avalanche hazard situation is presented to construct a picture of the potential threats. This paper provides information about avalanche fatalities and avalanche mitigation works in Turkey.

**Keywords**: Avalanche situation, avalanche hazard in Turkey, avalanche hazard management.



## 1 Introduction

Natural hazards are defined as potentially damaging processes resulting in movement of water, snow,
ice, debris, rock fall and landslides on the surface of the earth (Kienholz et. al, 2004; White and Haas,
1975). Of the natural hazards, an avalanche can be described as a falling mass of snow that may
contain ice, rock, or soil (Schweizer et al. 2003; McClung and Schaerer, 2006). Avalanches may not be
considered as a problem limited only to local inhabitants of mountainous areas. Avalanches can cause
serious damage to settlements, properties, and transportation facilities, and infrastructure such as
railways and main roads (Höller, 2007; Sauermoser, 2008; Holub and Fuchs, 2009; Simonson et al.
2010, Kurt, 2014).
Turkey is divided in to 81 provinces and these provinces are organized into 7 regions: the Marmara
region, the Black Sea region, the Aegean, the Mediterranean region, Central Anatolia, East Anatolia,
and Southeast Anatolia (Figure 1). The total population of Turkey is 78,6 million according to the 2016
estimate based on Turkish Statistical Institute. Mostly Eastern part of Turkey is mountainous region.
According to a recent study (Elibüyük and Yılmaz, 2010), the mean altitude of Turkey is 1141 m a.s.l.,
more than three times higher that of Europe (300 m a.s.l.), with a mean slope angle of 10◦. Altitudes
higher than 1500 m with slopes greater than 27◦ cover 5.1 % of the total area (Aydın et al., 2014).




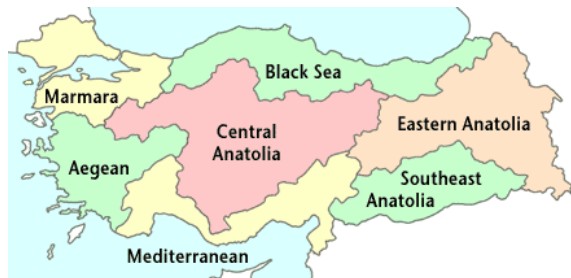


**Figure 1.** Turkey consists of 7 regions.
Northeast, east and southeast parts of Turkey are the most endangered areas based on
recorded avalanche events (Figure 2). Generated elevation and inclination map of
Turkey based on SRTM (Shuttle Radar Topography Mission) database and maximum
snow height map verify that higher inclination and altitude including heavy snowfall
trigger more avalanches in Turkey (Figure 3-4).

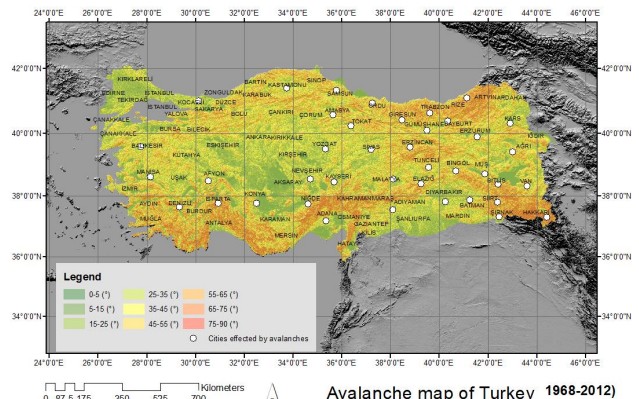


**Figure 2 Cities affected by avalanches between 1968 and 2012 (after TABB)**


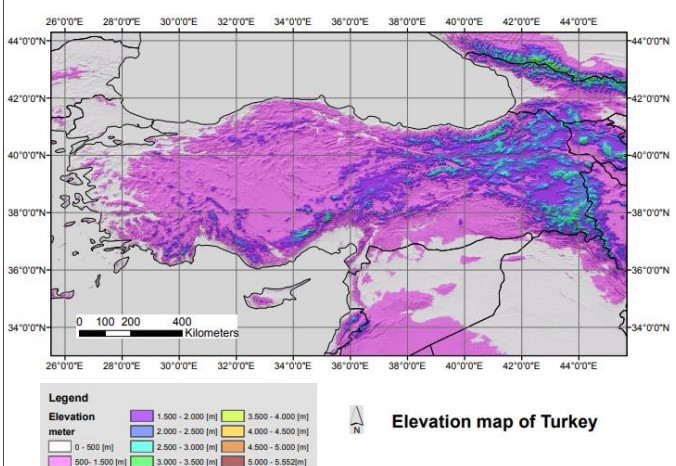


**Figure 3 Elevation map of Turkey**

57                    **Figure 4 Map of maximum snow heights in Turkey**


Turkey is known as a country with many summer holiday activities due to the seas that
surround three sides of the country and its historic places. However, avalanches also
constitute risk situations for settlements, inhabitants and winter sports centers.
Avalanche events have increased in Turkey, especially since the 1990s. For example,
during the winter of 1992, 158 avalanche events were recorded, 453 people were killed,
and 108 people suffered injuries in Turkey (Gürer 2002).  Such large events gave
impetus to avalanche control works. Authorities have become particularly aware of the




66 destruction caused by avalanches after the 1990s, and they have started to try to avoid

67 future avalanche damage (Table 1). Until today, different governmental bodies have

68 been dealt with avalanches to keep people and property in safe. For example; Turkish

69 State Railways (TCDD); General Directorate of Highways (KGM); General Directorate

70 of Forestry (OGM); General Directorate of Combating Desertification and Erosion

71 (ÇEM);Disaster and Emergency Management Presidency (AFAD) prepare avalanche

72 control projects. Mostly passive avalanche control methods are used instead of

73 permanent control methods due to absence of organized avalanche control service.

74

75    **Table 1 Organizations deal with avalanches in Turkey.**

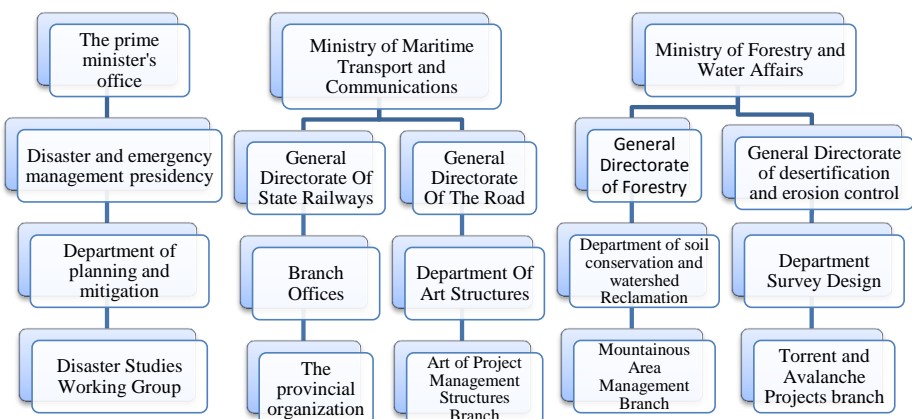

76

77 The objective of this study is to analyze the current situation in Turkey in terms of

78 avalanche hazards, avalanche mitigation works, and authorities by assessing the

79 avalanche risk influencing factors. Finally, it is aimed to describe avalanche situation in

80 Turkey.

81



## 2   Avalanche Hazards

### 2.1 Avalanche triggering factors

Topographic factors, vegetation factors and weather conditions such as terrain, meteorological, and snowpack factors may be very important in the occurrence of avalanches (Hendrix et al. 2005; Hageli and McClung, 2003; McClung and Schaerer, 2006; Schweizer et al. 2003; Höller, 2007). For example, avalanches generally occur on slopes between inclination 28° and 55° (McClung and Schaerer, 2006; Miklau and Sauermoser, 2011, Selçuk, 2013). Avalanches normally are not triggered on slopes outside this range because snow masses tend not to accumulate on such slopes (Sullivan et al. 2001).

Slope aspect can be particularly important factor due to snowmelt, solar radiation and wind loading on the snowpack (Grimsdottir and McClung, 2006; Cooperstein et al. 2004; McClung and Schaerer, 2006). For example, south-facing slopes in the northern hemisphere can be especially dangerous in the spring for occurrence of wet snow avalanche when heated by the sun (Dubayah, 1994). North-facing slopes may be slower to stabilize than slopes facing in other directions (Daffern, 2009). Another example, leeward slopes, slopes facing away from the wind, are dangerous because this is where the snow collects and may form an unstable slab (Meloysund et al. 2007). On the other hand, windward slopes that face the wind generally have less snow and are usually more stable (McClung and Schaerer, 2006; Dubayah, 1994). Moreover, because of solar radiation and wind-drifted snow, the strength and thickness of the snow cover and distribution of weak layers can vary with the aspect (Grimsdottir and McClung, 2006).



Statistical works indicated that; most avalanches fell in northern, northeastern and
eastern aspects, which are the lee and shady aspects (Grimsdottir and McClung, 2006).

"Snowpack forms from layers of snow that accumulates in geographic regions and high
altitudes where the climate includes cold weather for extended periods during the year"
(Broulidakis, 2013). Snowpack factors are snowpack depth and structure, such as
hardness, layering, crystal forms and free water content (McClung and Schaerer 2006).
According to Gaume et al. (2013), the spatial variability of snowpack properties has an
important impact on snow slope stability and thus on avalanche formation. For example,
as the snow falls it settles in layers of varying strength and weakness. Because
numerous layers constitute a snowpack, it is important to understand the properties of
each layer of the weak layer such as overlying load, densities, temperature gradient and
crystal types, because each one forms under varying weather conditions and will bond
to approaching layers differently (Jamieson and Johnston, 2001). Weak layers deep in
the snowpack can cause avalanches even if the surface layers are strong or well-bonded
(Jamieson et al. 2003).
Vegetation cover is among the factors affecting avalanches by increasing or decreasing
friction on the surface (Butler, 1972; Simonson et al. 2010). Smooth slopes with pasture
can accelerate formation of avalanches due to lack of resistance (Tunçel, 1990). If there
is vegetation cover in the form of shrubs in the avalanche release zones, the vegetation
cover can hold the snow mass and delay the initiation of avalanche. According to Brang
(2001) dense forests are also effective to reduce avalanches by preventing the avalanche
release in initiation zones. On the other hand, vegetation analysis can be used to survey
past avalanches and to estimate the frequency and intensity of snow-slide events for



specific avalanche path locations and time periods of interest (Burrows and Burrows,
1976; Carrara, 1979; Mears, 1992; Jenkins and Habertson, 2004; Casteller et al. 2007;
Bebi et al. 2009; Simonson et al. 2010).

Wind loading may occur without precipitation, by scouring of snow on exposed
windward slopes and subsequent deposition of this scoured snow on lee slopes
(McClung and Schaerer, 2006; Schweizer et al. 2003). Variations in wind speed and
snow drift can be important that they form layers of different density or harness creating
stress concentrations within the snowpack (McClung and Schaerer, 2006). For example,
it has been assumed that snow drift peaks at a wind speed of about 20-25 m s$^{-1}$   and
decreases with even higher wind speeds (Schweizer et al. 2003).

Temperature is a decisive factor contributing to avalanche formation, particularly in
situations without loading (Schweizer et al. 2003). The temperature of the weather and
of the snowpack has an increasing effect on the risk of avalanche. According to
McClung and Schaerer (2006), the mechanical properties of snow are highly
temperature dependent. In general, there are two important groups of competing effects:
metamorphism (depending on temperatures) and mechanical properties (excluding
metamorphism effects) including snow hardness, fracture propagation potential and
strength (Schweizer et al. 2003). The probability of powder snow avalanche is high in
the presence of low weather temperature and wind (McClung and Schaerer, 2006). On
the other hand, temperature rises in the spring can cause wet snow avalanches
(McClung and Schaerer, 2006; Tunçel, 1990). Another example regarding temperature
is solar radiation because it can prepare conditions for avalanches to initiate



(Grimsdottir and McClung, 2006). Solar radiation can present greater risk situations in
alpine regions than lower down, due to the open terrain (Grimsdottir and McClung,
2006; Cooperstein et al.; 2006). Surface hoar forms when relatively moist air over a
cold snow surface becomes oversaturated with respect to the snow surface, causing a
flux of water vapor, which condenses on the surface (McClung and Schaerer, 2006).

**2.2 Avalanche History of Turkey**
The earliest recorded avalanche fall in Turkey occurred in 1890 (Varol and Yavaş,
2006). Based on avalanche records from AFAD that consist of all types of
documentation (reports, photos etc.) from 1890 to 2014, 1997 avalanche events
occurred, and more than 1446 people were killed by avalanches (Figure 1). Based on
AFAD records, on average, avalanche events have caused 30 deaths in Turkey every
year as well as damage to villages, settlements, infrastructure and forests over the last
30 years (Figure 5).

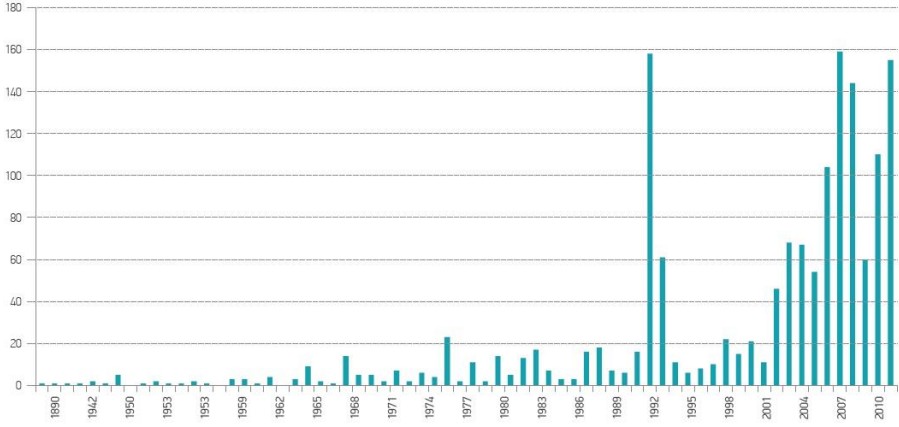



**Figure 5 Avalanche fatalities between 1890 and 2014 (AFAD)**





Avalanche observations of past and present avalanche activity are of the utmost
importance for any avalanche forecasting operation and avalanche control concepts
(Laternser and Schneebeli, 2002). According to Gürer (2002), prior to the 1950s
avalanche events were not recorded in Turkey by authorities unless they caused deaths
or injuries. In order to create a database for all disasters including avalanches, TABB
(Turkish National Disaster Archive) was established with support from AFAD (Disaster
and Emergency Management Presidency) in 2004.

In order to create a database on Turkey as a whole, including previous disasters,
Turkish National Disaster Archive (TABB) was established in 2004. Written
documents, reports, photos, and other types of information have been collected from the
following governmental bodies: AFAD, General Directorates of Food Control,
Gendarmerie General Commands, Police, Turkish Atomic Energy Authority and Media.
Then, some criteria imposed for records by TABB: the exact date if known, in the
absence of a certain date, the number of fatalities, in the absence of fatalities, the
number of injured.  Specifications are searched within these data parameters. However,
so far all the collected data has not been entered into TABB databank and the
monitoring process is behind schedule.

During the winter of 1992, a total of 158 avalanche events were recorded, 443 people
were killed, and 108 people suffered injuries in Turkey (Gürer, 2002; Gürer et al.,
1995). The distribution of several major avalanche events of 1992 and 1993 are shown
in Table 2. One reason for avalanche fatalities in Turkey was the heavy snowfall. For
example, the heavy snowfall (Figure 6) in January and February caused many avalanche
events in eastern Turkey (Gürer, 2002). Many main roads were closed and many
villages were affected by avalanches.







**Table 2 Major avalanche events of 1992 and 1993 in Turkey**

| Date | Deaths | Details |
| --- | --- | --- |
| 02.01.1992 | 20 | Due to heavy snowfall, 20 inhabitants were killed in *Karabeya* village, *Yüksekova*, province of Hakkari. |
| 21.01.1992 | 10 | 10 inhabitants were killed in *Kesmetaş* Village, *Şirvan*, province of Siirt. |
| 01.02.1992 | 97 | 71 soldiers in Turkish army and 26 inhabitants in the *Görmeç* village were killed due to avalanche fall in province of Siirt. |
| 07.02.1992 | 55 | 31 person in *Boğazören* village, *Beytüşşebap*, province of Şırnak; 13 people in different villages in province of Batman, 5 people in *Erimli*, province of Elazığ, 6 people were died in province of Bingöl and Diyarbakır. |
| 08.02.1992 | 21 | 15 inhabitants in *Çığlıca* village, province of Şırnak, 6 inhabitants were killed in *Tatlıca*, province of Batman |
| 21.02.1992 | 32 | 32 soldiers were killed in both *Eruh* and *Uludere*, province of Siirt |
| 25.02.1992 | 26 | Due to heavy snowfall, 26 inhabitants in *Anaköy* Village, *Gevaş*, province of Van. |
| 18.01.1993 | 59 | 59 inhabitants were killed and 21 were injured due to avalanche event in *Üzengili* village, province of Bayburt |
| 25.02.1993 | 26 | 26 inhabitants were killed in Anaköy, province of Van |
| 27.02.1993 | 6 | 6 passengers were killed on the Hakkari-Van main road due to avalanche event |







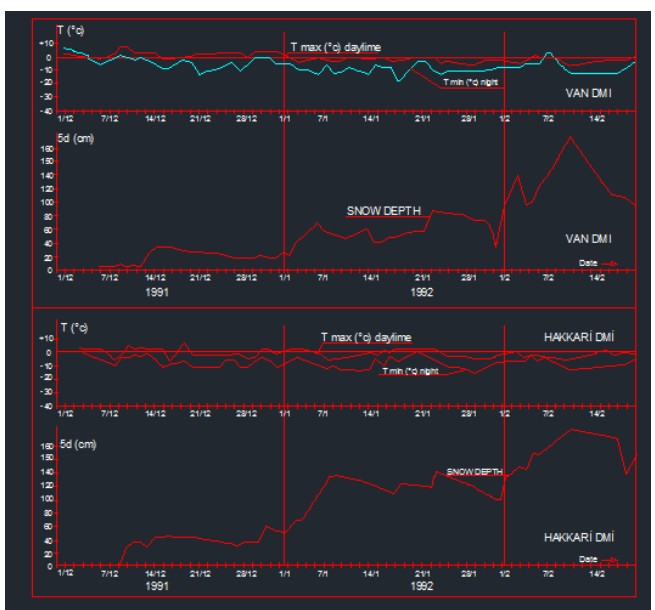

**Figure 6. Snow depths, maximum and minimum weather temperature in the winter of 1991-1992,**
**in Hakkari and Van Province (Gürer, 2002).**

One of the most deadly avalanches in Turkey occurred in the winter of 1992. A brigade
of 71 Turkish soldiers and 26 inhabitants were killed by an avalanche in the village of
*Görmeç*, province of Siirt on 01.02.1992. According to Borhan and Kadıoğlu (1992),
the day the Görmeç avalanche happened, the weather was rainy and snowy. Another
detail of this avalanche that contributed to the number of deaths was that there was no
avalanche control structure in the region.

Another deadly avalanche occurred in the village of *Üzengili,* Bayburt Province, on
January 18th, 1993, at 07:45 am. This avalanche killed 59 people and destroyed 62
buildings. According to Taştekin (2003), on  January 16th, the weather temperature was
− 5.0 °C, the next day air the temperature dropped by 10 degrees and became -15°C (Figure
5). As a result, the snow surface cooled.  During the daytime of January 17th, the air
temperature increased. Snowfall during the night of January 17th, (between 21:00-07:00)
(Figure 7-8). New snow precipitation caused an additional load and the previous snow layer
could not carry new snow, and in the morning of the 18th, the Üzengili avalanche occurred.



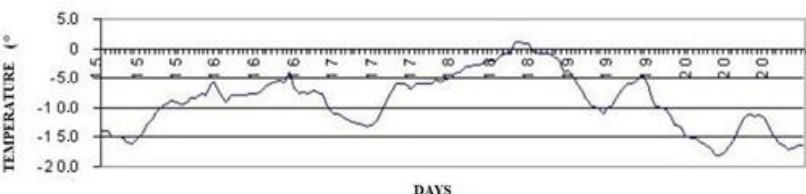


**Figure 7 Air temperature of Bayburt Üzengili in 18.01.1993 (Taştekin, 2003)**


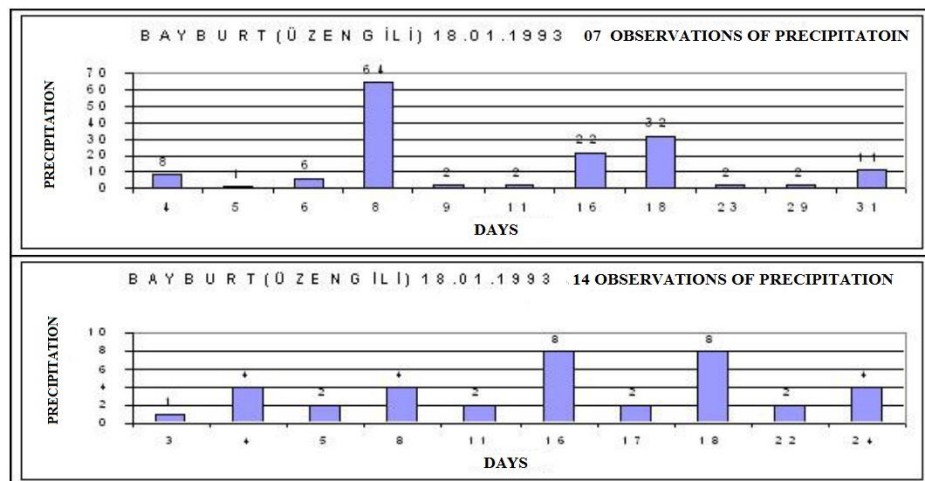



**Figure 8 According to 7 and 14 of records, precipitation in Bayburt, Üzengili (Taştekin, 2003)**


The Palandoken ski resort is located on the northern slopes of the Palandoken range in
Erzurum province; skiing is possible for 150 days in a year, skiing altitude is 2200-3176
m.  So far, 9  skiiers lost their lives  between 1996-2006 and three more skiers in total
(Figure 9).


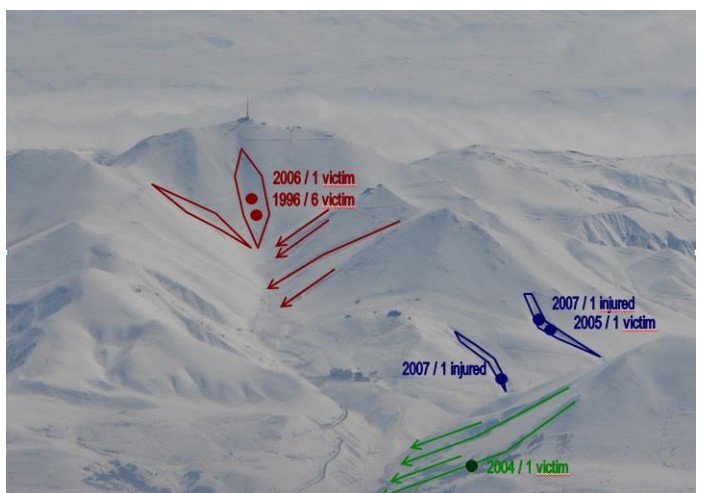


**Figure 9 Avalanche locations in Palandöken skiing centre in Erzurum province.**


## 3   Avalanche Protection

Today governmental bodies have been dealing with avalanche control issue, especially over the past few decades, in order to control avalanches and keep people and their property safe in Turkey such as the Ministry of Forestry and Water Affairs, *General Directorate of Forestry (OGM); General Directorate of Combating Desertification and Erosion (ÇEM);* Ministry of Interior, *Disaster and Emergency Management Presidency (AFAD); General Directorate of Highways (KGM).*

247

### 3.1 Avalanche mitigation projects in settlement areas in Turkey

If an avalanche occurs in an area of no settlements, no property, or no traffic, it does not constitute risk (Holub and Fuchs, 2009). Hence, avalanche protection may not be necessary in these uninhabited areas. On the other hand, if an avalanche presents hazard, a decision has to be made quickly to ensure maximum safety of endangered objects in the hazardous zone. So, avalanche protection works reduce the hazard avalanches pose to human lives and properties. Today, numerous endangered settlements still have no protection in Turkey.




Avalanche protection may be divided into temporary and permanent measures
(McClung and Schaerer, 2006). Temporary measures are applied for short periods when
avalanches are expected to occur. On the other hand, permanent measures usually
require expense for engineering works but perform without the need for a daily hazard
evaluation (McClung and Schaerer, 2006). There are very few snow pack supporting
structures or plans to protect settlements in Turkey beyond the current method of
reforestation.

In the starting zones, avalanche control methods are implemented to prevent the start of
avalanches or limit the snowpack motion that can be triggered by snow movement due
to steep slopes, with the help of supporting structures, snow fences or other type of
apparatus (McClung and Schaerer, 2006; Höller, 2007; Margreth et al. 2007). The
Uzungöl project was the first recorded avalanche-control project implemented in 2004
by OGM (General Directorate of Forest). This project comprised 6680 meters of steel
snow fences, 3340 meters of snow breakers, and mini-piles constructed in the avalanche
release zones (Figures 10-11).

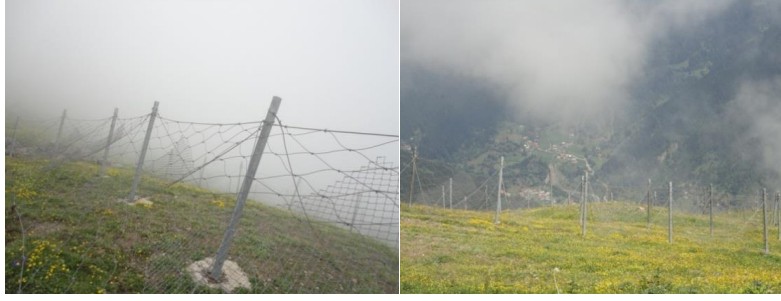

**Figure 10 Snow nets used as supporting structures in release zone, Uzungöl, province of Trabzon.**




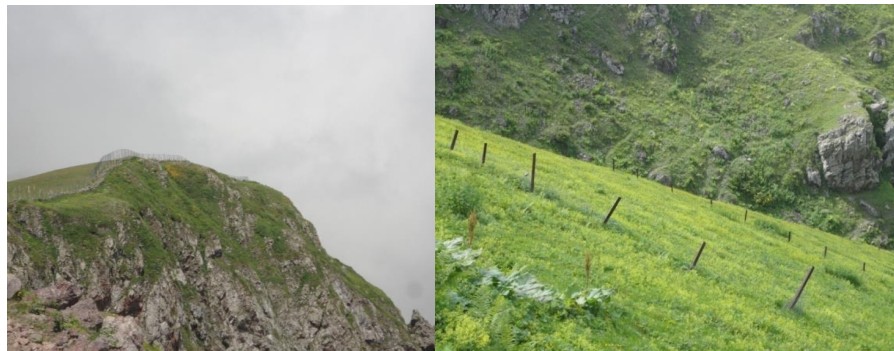



**Figure 11 Snow fences and mini-piles  in Uzungöl, Trabzon (Photo: T.Kurt, 2012**


In recently, steel snow bridges have been constructed also in Çaykara, Karaçam, in
Trabzon Province in 2016 (Figure 12).

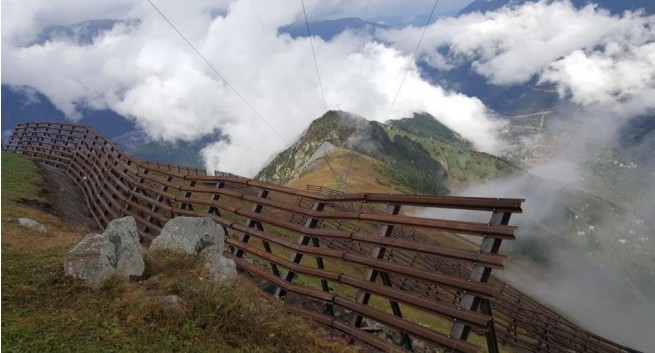


**Figure 12 Steel snow bridges in Trabzon (Photo: Anonym).**


**3.2 Avalanche hazard zoning and mapping in Turkey**
Avalanche hazard maps can give an idea of the safety level of a certain area in regard to
the risk of natural disaster (avalanches, rock falls, or torrents) (Holub and Fuchs,
2009).Avalanche hazard zoning is used in Turkey to prevent buildings being
constructed in areas endangered by avalanches and to indicate avalanche prone areas.





Three different risk levels are used for avalanche hazard maps (Varol and Şahin, 2006)
(Table 3).
**Table 3 Types of avalanche levels used in Turkey (after Varol and Şahin, 2006)**

| Avalanche Zone | Introduction |
|---|---|
| Red | High risk level. No construction is allowed and unsuitable area for residence. |
| Blue | Middle avalanche risk level, permanent use for settlement and infrastructure is possible but with additional safety measures |
| White | No avalanche risk |


On the other hand, avalanche hazard zonings are marked on maps for all avalanche
prone areas that are designated as avalanche paths (avalanche areas), active avalanche
paths, potential avalanche flow tracks, and possible avalanche flow tracks (Figure 13).
These maps are drawn based on avalanche chronology, topographic properties (aspect,
slope etc.), and vegetation cover.

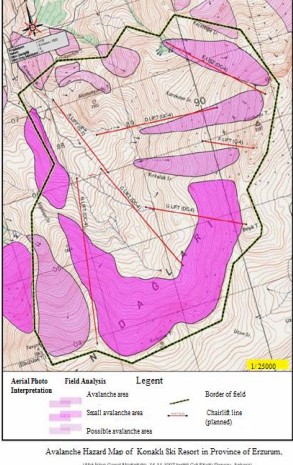


**Figure 13 Avalanche hazard zoning in Turkey.**



### 3.3 Avalanche control in winter resorts


Most common avalanche control methods for winter resorts in the regions affected are
avalanche forecasting, control programs, closure of ski paths and warning signs at
defined locations. Also, in the event of heavy snowfall, methods in use include artificial
avalanche release by using the Gazex system (Figure 14) under controlled conditions in
order to trigger smaller, less-destructive avalanches, closure of avalanche paths. For
example, the Gazex system is being used in the Erzurum Palandöken ski resort.

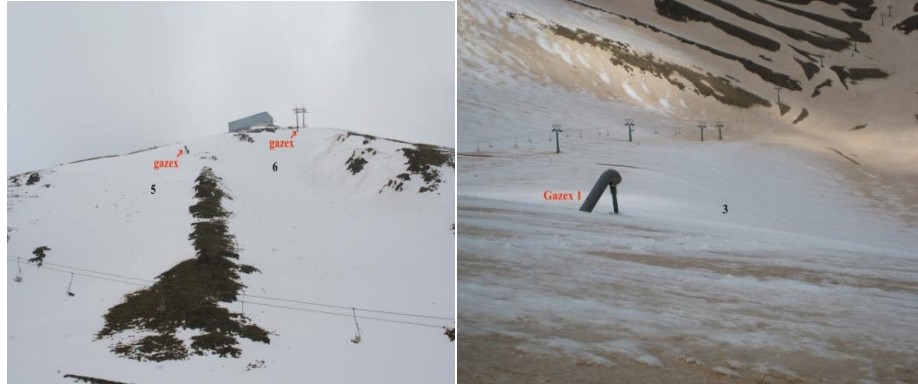


**Figure 14 Artificial avalanche release by using Gazex, Palandöken, Erzurum Province (Photos:**
**Anonymous)**

### 3.4 Avalanche Mitigation near Main roads


Avalanche hazards can be reduced by defense structures such as avalanche galleries
(Figure 14), snow sheds or snow breakers (Figures 15-16), and closure of main roads.
KGM is responsible for construction and safety of roads in Turkey. In this context,
KGM takes measures against avalanches by constructing avalanche galleries and snow
sheds (Figure 17-18)






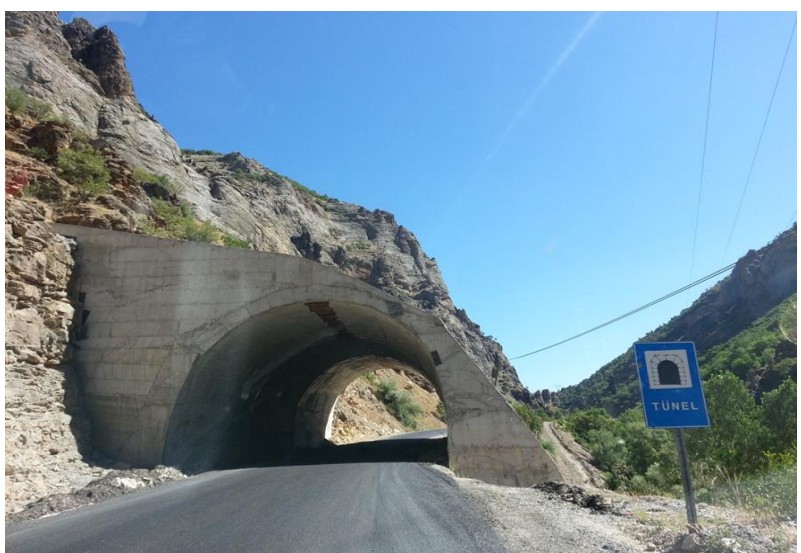

**Figure 15 Avalanche gallery (28 meter long) in Elazığ Province (Photo: Anonymous)**


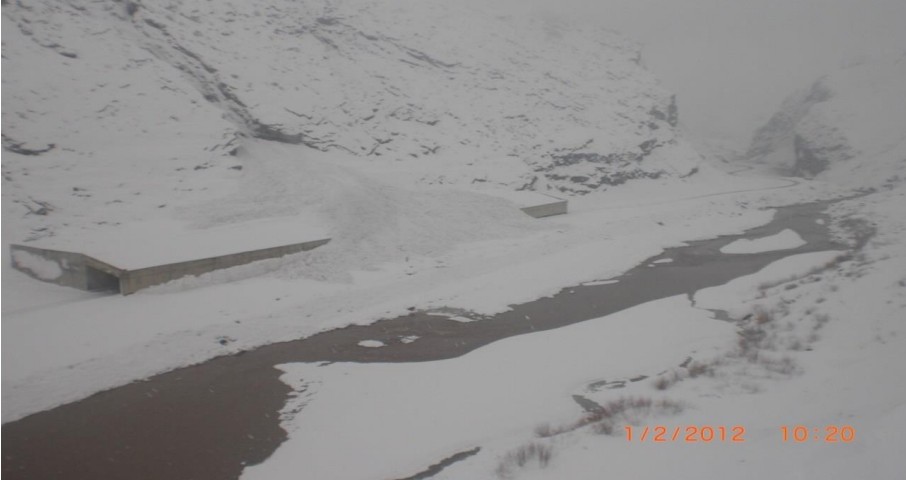

**Figure 16 Snow shed (206 meter long) in Province of Hakkari  (Photo: Anonymous)**



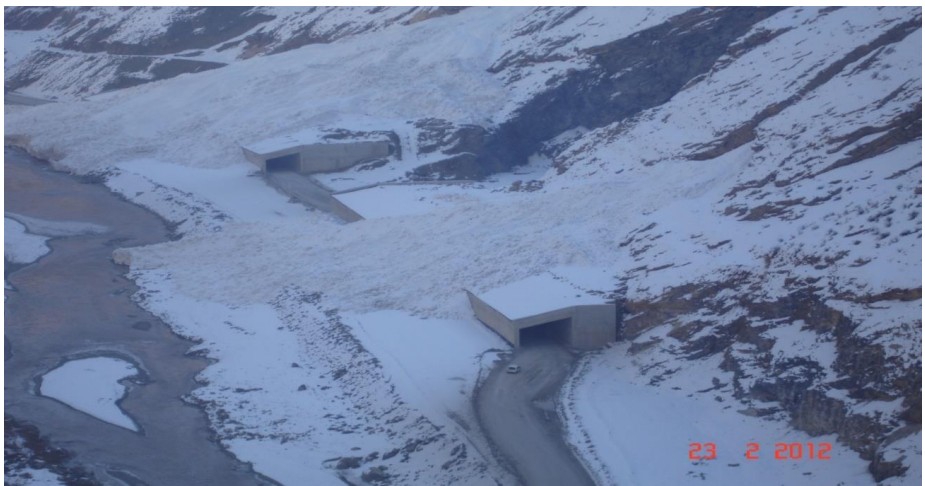


**Figure 17 Effectiveness of snow sheds built by KGM in Turkey (Photo: Anonymous)**


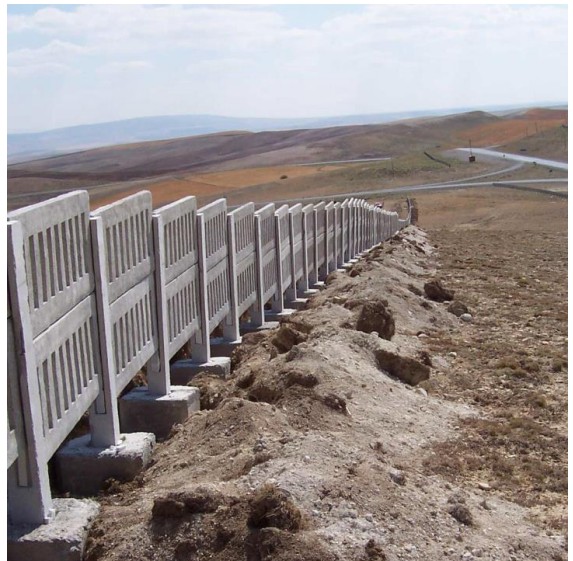


**Figure 18 Snow breakers in order to prevent snow-drift built by KGM (Photo: Okur, 2008).**


## 4 Conclusion and Discussions

In avalanche control, the first required data is the snowpack-related records. Turkish
meteorological stations were generally established before in or around the cities.
Therefore, there is lack of records regarding starting zones with respect to snowpack



properties such as density, snow depths etc. The Government has tried to close this gap
but the speed should be increased. According to Höller (2007), great avalanches may be
released during storm periods when the accumulated new snow that has fallen within
three days is more than 80 cm. The number of automated weather stations (AWOS)
should be increased in the Turkey thus enabling measurements of new three-day snow,
snow density, depth and temperature, wind direction, wind velocity, precipitation and
other data in areas prone to avalanches.
Turkey lacks an avalanche archive system that records avalanches and related
information, producing statistical analyses etc. Due to its special criteria (i.e. dependent
on a report of avalanche) the TABB may not include all avalanches. For example,
according to the TABB records, the earliest recorded avalanche in Turkey was on
01.01.1968 in the province of Elazığ. However, Varol and Yavaş (2006) reported that
the earliest recorded avalanche occurred in 1890. This clearly shows that the TABB
project should be completed as soon as possible, which should consist of written
documents including the date, location, and other relevant details of avalanche events.
In addition, the TABB records should be kept up to date.
The second step in avalanche control is preparing the risk maps There are risk maps
prepared and a few in the process of being prepared but if one takes into account the
number of potential avalanche areas, much remains to be done.
Turkey should create its own avalanche guidelines to determine technical standards
guide and reference that establish the details for creating and upholding an effective
avalanche control.  Avalanche hazard mapping may be improved by using various
levels to indicate risk levels in terms of impact pressure of the snow mass.
According to Li (1998), new generation high-resolution satellite images will provide
strong geometric capabilities. So, required departments (ÇEM, OGM, KGM, AFAD)
should specialize and produce precise maps (large-scale maps) such as those created
using airborne laser scanners or terrestrial laser scanners in order to increase accuracy of
topography analysis (i.e. slope, aspect analysis, avalanche modelling).



Turkish authorities are now keen to avoid future avalanche damage further to large
avalanches over the last two decades. Training for avalanche control has gained
importance and some forest engineers have been sent abroad by faculties of forestry and
the Forest Ministry. The ministry ordered several avalanche-control projects with the
support of forestry faculties. Implementation of these projects, the first steel snow
bridges implemented in 2016.

After the Üzengili avalanche that occurred in 1992 in the Bayburt province, authorities
decided to evacuated these villages:  Üzengili, Yaylapınar, Kavlatan, Harmanözü and
Dumlu. These villages (889 persons) were relocated to safe-areas (Report, 2011).

Some avalanche control projets  are being implemented by governmental bodies and the
private sector. For example, the OGM of the forest ministry has been implementing
snow nets, and micro piles since the early 2000s in Trabzon, KGM have been
constructing snow galleries and snow fences in eastern Turkey and artificial avalanche
release is being used in ski resorts in the private sector. In addition to these precautions,
some hotels have forbidden skiing in areas at risk. The areas at risk are marked by
warning signs.

In Turkey there are a number of governmental bodies responsible for avalanche
protection (i.e., AFAD, OGM, ÇEM, and KGM), and this multi-department situation
could cause uncertainties. There should be a single organization responsible for
avalanches to prevent complexity in making risk maps, preparing avalanche control
projects, and implementing projects similar to the Austrian model. Aıustria has the
Austrian Service for Torrent and Avalanche Control (Die Wildbach und
Lawinenverbauung), which is part of the forest department.  This office only deals with
avalanches and torrents, and arranging projects for avalanche control measures with
their specialist staff.





**Acknowledgements**
This work was funded by Scientific Research Projects Units of Istanbul University
Project number: 47628. The present work partly contains some results presented in the
doctoral dissertation completed by Tayfun Kurt under the advisement of Prof Dr.
Hüseyin Emrulllah Çelik at the Department of Forest Construction and Transportation
at the Faculty of Forestry, Istanbul University. The practical study, which lasted a few
months, was performed at the Rize,Erzurum, Isparta Province. I wish to express my
special thanks to Prof. Dr. Hüseyin Emrullah Çelik, Mr. İsmail Bulut from ÇEM, Mr.
Ali Haydar Görmez from KGM and to Mrs. Belgin Baran, from AFAD, for providing
information on respectively Erzurum Paladandöken, avalanche control in KGM and
avalanches in Turkey.

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
