# Peer review of "Assessment of avalanche hazard situation in Turkey during years 2010s"

_Natural Hazards and Earth System Sciences, 2018_

## Referee Comment (RC1) · Anonymous Referee #1 · 4 Sep 2018

**Review of the manuscript „Assessment of avalanche hazard situation in Turkey during years 2010s"  (Kurt, 2018)**

The manuscript describes the Turkish approach of mitigation work against snow avalanches. It characterizes the topographic and meteorological features of Turkey. Several examples of catastrophic snow avalanches are listed and a historical review is given with the time series of fatalities since 1890. The basics of hazard mapping in Turkey are pointed out and some protection measures are shown.

In principle the theme can be of interest for the readers of NHESS. However, the manuscript does not fulfill scientific standards. It requires a revised structure and a reformulation of major parts of the text. Additionally, the source of several photos is unknown. Therefore, I suggest to reject the manuscript for publication in this form.

The statements in the *general comments* explain this in more detail. Furthermore, the manuscript contains numerous inaccurate formulations and errors which are listed in the *specific comments and technical corrections*.

**General comments**

The manuscript contains several photos which illustrate the functionality of defense structures. In some cases, the sources of the photos are "anonymous". This implies that no permission for publishing can be provided. Maybe the copyrights of the images belong to other journals? Furthermore, sometimes the date of the photography is missing.

A scientific article has a pre-defined structure which is often called IMRaD-concept. IMRaD is an acronym for Introduction – Method – Results – and – Discussion. The NHESS journal does not presuppose this structure. However, it is a wise approach to prepare the manuscript in that way. Almost all articles of NHESS have this structure (excepted review articles, …).

The objectives of the study are announced at the end of the introduction (which is the right location). It is stated that the aims are "to describe" and "to analyze" the snow avalanche situation in the Turkey. This is neither a scientific question nor a hypothesis which have to be tested or verified. One approach to improve the manuscript could be to compare methods of hazard mapping and design of defense structures in Turkey with methods used in other countries. Why are there differences between guidelines? The introduction already indicates some differences to the European topography. This idea is just a suggestion!

The manuscript contains numerous typing errors and the wording is poor. The English language would deserve an additional check.

The acknowledgments indicate that the project was funded by the University of Istanbul and the author made his dissertation in this context. Why does the author call himself as independent researcher in the affiliation section (between title and abstract)?

**Specific comments and technical corrections**

Line 2 and the whole manuscript: replace "avalanche" with "snow avalanche"

Line 8: "risky situation especially for mountainous areas" is imprecise. Snow avalanches constitute risks against human or infrastructure. Reformulate the sentence.

Line 9: remove "have"

Line 10: replace "... 30 people per year ..." with "... in average 30 people per year …"

Lines 10-12: The Abstract should not include an example which was not shown in the text.

Line 14: replace "This research is focused on, known fatal avalanches …" with "This study is focused on fatal avalanches …", because it impossible to investigate unknown events.

Line 14: replace "obtanied" with "obtained" and remove the extra space at the beginning of the sentence.

Line 15: replace "... provides are reliable ..." with "... provides a reliable …"

Line 15: replace "contstitute" with "constitute"

Line 16: replace "situation" with "situations"

Line 16: remove the extra space in "... new  avalanche …"

Line 30: replace "et. al," with "et al.,"

Line 31: remove extra space at the beginning of the sentence

Line 32: add a comma between "Schweizer et al." and "2003"

Lines 34-35: reformulate the list, because transport facilities and infrastructures such as railways and main roads means the same; use "or" instead of "and"; "properties" is unclear.

Lines 35-36: add a comma between "Simonson et al." and "2010"

Line 36: replace the comma with a semicolon between "2010" and "Kurt"

Line 39: remove the extra space between "million" and "according"

Line 39: replace "78,6" with 78.6"

Lines 39-40: The statement requires a citation.

Line 40: Remove the extra space at the beginning of the sentence.

Line 43: add "a.s.l." after "1500 m"

Line 43: replace "greater" with "steeper"

Line 49: The Shuttle Radar Topography Mission is not cited adequate.

Lines 48-51: reformulate this sentence. Maybe it make sense to split the information in two sentences. Add "Fig. 2" (line 51), because the inclination map is mentioned.

Lines 52-53: This is the inclination map. However this is neither referred in the figure caption nor in the legend.

Lines 52-57 (Fig.2,3,4): use the same layout for all figures (grid lines, scale, etc.). If the author did not exclusively create the Fig. 3 for this article, the figure must be cited with permission of the owner.

Line 62: replace "Avalanche events have increased …" with "The number of avalanche events increased …"

Line 64: remove the extra space at the beginning of the sentence

Lines 64-65: replace "... gave impetus ..." … with "stimulated ... ".

Lines 65-67: reformulate the sentence. It is unclear if it was just tried or if something was actually changed.

Line 67: remove the extra space at the beginning of the sentence

Line 71: add a space before "Disaster …".

Lines 77-80: The objectives of the study are unclear.

Line 84: replace "Topographic factors, vegetation factors and …" with "Topographic properties, vegetation characteristics and …".

Line 86: wrong citation, replace "Hageli" with "Haegeli"; see also in the references (line 462).

Line 86: wrong year of citation, replace "Hageli and McClung, 2003" with "Haegeli and McClung, 2007".

Line 87: add comma between "Schweizer et al." and "2003".

Line 89: replace comma with semicolon after "Sauermoser, 2011".

Line 91: add comma between "Sullivan et al." and "2001".

Line 93: replace "Slope aspect … " with "The slope aspect …".

Line 94: add comma between "Cooperstein et al." and "2004".

Line 96: replace "... can be especially dangerous in the spring …" with "... can especially be dangerous in spring …".

Line 100: replace "collects" with "accumulates".

Line 100: add comma between "Meloysund et al." and "2007".

Line 105: remove the semicolon.

Line 110: remove the extra space at the beginning of the setnence.

Line 110: replace "snowpack depth" with "snow depth".

Line 111: add a comma between "McClung and Schaerer" and "2006".

Line 116: remove extra space between "such as" and "overlying".

Line 116: replace "gradient" with "gradients".

Line 120: add a comma between "Jamieson et al." and "2003".

Lines 121-122: reformulate the sentence. Suggestion: "Among other factors the vegetation affects the friction …"

Line 122: add a comma between "Simonson et al." and "2010".

Line 124: replace "is vegetation" with " is a vegetation".

Line 125: replace "avalanche" with "avalanches".

Line 130: add a comma between "Casteller et al." and "2007".

Line 131: add a comma between "Bebi et al." and "2009".

Line 131: add a comma between "Simonson et al." and "2010".

Line 135: add a comma between "Schweizer et al." and "2003".

Line 139: add a comma between "Schweizer et al." and "2003".

Line 142: add a comma between "Schweizer et al." and "2003".

Lines 142-143: reformulate the sentence and precise the statement. Use "air temperature" instead of "temperature of the weather".

Line 148: add a comma between "Schweizer et al." and "2003".

Line 148: replace "The probability of powder snow avalanche …" with "The probability of the formation of powder snow avalanches …".

Line 149: replace "weather temperature" with "air temperature".

Line 149: replace "and wind" with "and high wind speed".

Line 150: replace "temperature rises in the spring" with "rising temperatures in spring".

Line 151: replace "Another example regarding temperature is …" with "Another factor influencing the energy budget is …".

Line 152: replace "prepare" with "create".

Line 152: replace "avalanches to initiate" with "avalanche formation".

Lines 153-155: reformulate the sentence.

Line 155: citation error: replace "Cooperstein et al., 2006" with "Cooperstein et al., 2004".

Line 163: replace "Figure 1" with "Figure 5".

Line 187: replace "databank" with "database".

Line 188: replace "... is behind schedule." with "... is behind the schedule.".

Line 192: replace "of 1992" with "in 1992".

Line 202: replace "of 1992" with "in 1992".

Line 202: replace "people were died in" with "people died in".

Line 202: replace "in both Eruh and Uludere" with "in Eruh and Uludere".

Line 202: remove two blank lines in the table.

Line 202: remove extra spaces on 25.02.1993 and 27.02.1993.

Line 202: Please add the region of the locations. The regions were already introduced in figure 1. That supports the reader, who is not from Turkey.

Lines 212-213: replace ", the day" with ", when the day".

Line 215: replace "control structure" with "defense structure".

Line 218: the date and time format changed in the manuscript (see line 212 and table 2 and the whole document).

Line 219: replace "weather temperature" with "air temperature".

Line 220: mixed format: minus symbol, space between numbers and units.

Line 220-221: replace "Figure 5" with "Figure 7".

Line 221: remove the extra space before "During the daytime".

Lines 223-224: reformulate the sentence.

Lines 225-226: replace "18.01.1993" with "from 15th January 1993 to 20th January 1993".

Lines 228-230: The figure is unclear: missing units; missing days; "07" and "14 observations of precipitation", it sounds like "a number of observations" or is it the time of recording?

Figure 6, 7, 8: the layout is not equal and the figures are cited. However, no comment like "with permission" is given.

Line 233: add "a.s.l." after "m".

Line 234: remove extra space between "lives" and "between".

Line 265: replace "start" with "release".

Line 266: replace "limit" with "inhibit".

Line 268: add a comma between "Margreth at al." and "2007".

Lines 274-275: date is missing, source/owner is not given and permission for publishing is missing.

Lines 278-280: date is missing.

Line 280: add space between "T." and "Kurt".

Line 280: ")" at the and of the line.

Line 282: replace "In recently" with "Recently".

Lines 285-286: date, owner and permission are missing.

Line 291: add space between "2009)" and "Avalanche".

Line 303: add "Example of …"

Line 304: replace "winter resorts" with "ski resorts".

Lines 305-307: reformulate the sentence.

Line 307: replace "defined" with "pre-defined".

Line 308: "Gazex system" requires a citation.

Line 312-314:  date, owner and permission are missing.

Line 317: replace "Figure 14" with "Figure 15"

Line 317: replace "Figure 15-16" with "Figure 16-18"

Line 320: replace " (Figure 17-18)" with "."

Lines 321-322: date, owner and permission are missing.

Lines 324-325: owner and permission are missing.

Lines 326-327: owner and permission are missing.

Lines 329-330: date is missing.

Line 330: replace "prevent snow-drift" with "influence the snow deposition".

Line 332: replace "Conclusion and Discussions" with "Discussion and Conclusions".

Line 334: replace "before in or around the cities" with "close to cities".

Line 336: replace "The Government has tried ... " with "The government tried …"

Lines 336-337: add a comma before "but"

Line 337: reformulate the statement. Suggestion: "... however, it is still in progress."

Line 339: replace "AWOS" with "AWS"

Line 340: replace "three-day snow" with "new snow sum (3 days)". If the AWS measures the snow depth the term "3 day snow depth difference" have to be used, because the settlement of the snowpack is included in the measurements.

Line 341: It is not common to measure the snow density with AWS. However, it is possible using e.g snow pillows. Provide information how snow density is measured with AWS in Turkey.

Line 341: replace "temperature" with "air temperature".

Line 341: replace "wind direction, wind velocity" with "wind speed and direction"

Line 342: replace "avalanches." with "avalanche release."

Line 354: replace "is preparing the risk maps" with "is to complete the risk maps in Turkey."

Line 360: remove extra space before "Avalanche hazards"

Line 403: replace "Prof" with "Prof."

Line 404: remove extra space before "at the Department"

Line 405: remove extra space before "The practical study"

Line 406: add space between "Rize," and "Erzurum"

Lines 413-546: The references are not ordered alphabetically.

Lines 448-449: not cited in the text.

Line 462: replace "Haeli" with "Haegeli"

Line 475: replace "assesment" with "assessment"

Line 476: replace "Journal of Journal of" with "Journal of"

Lines 480-482: not cited in the text.

Line 492-495: not cited in the text.

Line 497: add space between "Seattle," and "2006"

Lines 525-527: not cited in the text.

---

## Author Comment (AC1) · 9 Sep 2018

Dear **Anonymous Referee #1**

Your comment was upload into system to "Assessment of avalanche hazard situation in Turkey during years 2010s".   Thank you.

A few statements in the *general replies* explain your  a few comments in more detail.

**Statement 1**: The manuscript contains several photos which illustrate the functionality of defense structures.
**Reply** 1: The manuscript contains relevant photos for topic by topic. For instance; under the topic "Avalanche mitigation projects in settlement areas in Turkey**",** relevant photos were published as preventive a structure (snow fences, steel snow bridges) which means photos are not defense structures.

**Statement** 2: In some cases, the sources of the photos are "anonymous". This implies that no permission for publishing can be provided. Maybe the copyrights of the images belong to other journals? Furthermore, sometimes the date of the photography is missing.

**Reply** 2: As far as I understand, you mainly proposed to reject manuscript due to the anonymous expression of photos.  According to scientific ethic rules, it is not allowed to use someone photos without affiliation/ citation.  It is obviously known. I mean with expression of anonymous for photos in this paper; date of photo and photographer (the person) is not known clearly.
For instance, Figure 16, Figure 17, Figure 18. They are structures about Avalanche Mitigation near Main roads in Turkey. I attended a presenter/ speaker in a avalanche workshop which was organized by the Turkish Forestry Ministry in 2012. After my presentation, regional Chief of General Directorate of Highways who is Mr. Ali Haydar Görmez present his presentation and showed us these photos. I was planning to write a kind of paper during that times, therefore I took a permission to use these photos. (Also, he did not know the photographer and actual date of the phot. Therefore, I did not write anything about date and person. Please see acknowledges and name of Mr. Ali Haydar Görmez ). Also you can have a look that website: Please click

So, I got the permission to use photos which are not taken by myself. Copyrights of the images do not belong to other journals. These kind of photos are in format of JPEG format and they have 3-4 MB resolution dimension.  One more detail, I declared earlier that: The present work partly contains some results presented in the doctoral dissertation completed by my own self. Of course, I will write more detail for photos next time (i.e. date, person ).

**Statament 3**:  The acknowledgments indicate that the project was funded by the University of Istanbul and the author made his dissertation in this context. Why does the author call himself as independent researcher in the affiliation section (between title and abstract)?

**Reply 3**: Project was funded by the University of Istanbul. During that time I was working as research assistant at Istanbul Unıversity. But I have not work there since 01.09.2016.. Therefore, I wrote my address as independent researcher.

If I got to change to revise my paper, I will take into consideration your specific comments and technical corrections in order to improve this work. Thank you.

With my best regards,
Tayfun Kurt

---

## Referee Comment (RC2) · Anonymous Referee #2 · 10 Sep 2018

**Review Report of the Manuscript of Kurt, T (2018) (doi.org/10.5194/nhess-2018-205)**

**"Assessment of avalanche hazard situation in Turkey during years 2010s"**

Manuscript related with factors influencing avalanche formation , some fatality statistics and recent mitigation works of Turkey. Despite manuscript could interest the reader of NHESS, doesn't fulfill basic scientific standards. Contradiction between title and content. Very bad structured manuscript body. Unfortunately, manuscript not very well elaborated and contains some wrong information. I have suggest to reject it.

**General Comments**

Since title claim to assess avalanche hazard situation of Turkey for years 2010s reader expect data for given period. However fatalities statistics for last 30 years or for last 124 years from incomplete database. Reader couldn't find any data from 2010s. Who dies in Turkish avalanches? is unanswered question. Besides, no analysis for fatality numbers also from 2010s.

Avalanche triggering factors explained in more than three pages which is already known very well (Page 6-9). Doesn't contain any new information and no need to give such detailed information.

Table 1 shows organisational structure of avalanche related institutions in Turkey. But an up to date necessary. Since Turkey shifting to presidential system "The Prime Minister's Office", "Ministry of Maritime Transport and Communications" and " Ministry of Forestry and Water Affairs" reorganised and their names changed.

I couldn't understand why author present Table 2. Readers will expect info for years 2010s not some 20 years earlier. Similarly, Figure 6 related with snow depth info of an avalanche occured in 1991-92 which completely unnecessary. Same comment valid for Figure 7-9. Because author try to give insight for avalanche situation of 2010s but discuss in detail (which is not necessary) around one decade earlier events.

Avalanche hazard zoning and mapping in Turkey section also seems to be confusing (Page 16, Line 288) . In Turkey there is no guideline or any other article related for avalanche hazard zoning or mapping. The reference author cited just a proposal inspired from international references. Furthermore, Figure 13 not "hazard zone map" but a kind of "indication map" prepared based on aerial photo interpretation and field observation.

Some information also seems to be wrong in manuscript. For instance, Figure 5 shows avalanche fatalities in Turkey for years 1890-2014. According to Figure 5 in year of 2007 around 160, in year of 2011 around 150 fatalities occured. A small investigation on internet and TABB database doesn't proof Table 5. Furthermore in year of 1992 total 443 people were killed (page 10 line 190-191) due to avalanches but in Figure 5 we couldn't see it.

Manuscript contains many typing errors and not complete sentences. A native english speaker check recommended.

**Specific Comments**

Same with RC1. No need to repeat.

---

## Author Comment (AC2) · 30 Sep 2018

Dear **Anonymous Referee #2**

Your comment was upload into system to "Assessment of avalanche hazard situation in Turkey during years 2010s". However, while reading your comments, I was really struggling to understand. Because your English is not clear enough to understand for a reviewer.

A few grammar mistakes were written below.

False: I have suggest to reject it.
True: I have TO suggest to reject it.

False: Unfortunately, manuscript not very well elaborated..
True: Unfortunately, manuscript DOES not elaborated well.

False: Contradiction between title and content
True: THERE IS contradiction between title and content.

Also the paragraph which you wrote, it is really not clear understand and it contains grammar mistakes too.

"*Since title claim to assess avalanche hazard situation of Turkey for years 2010s reader expect data for given period. However fatalities statistics for last 30 years or for last 124 years from incomplete database. Reader couldn't find any data from 2010s. Who dies in Turkish avalanches? is unanswered question. Besides, no analysis for fatality numbers also from 2010s*".

I have to write that I'm sensing a little hostility in your comments. Because your comments do not fulfill scientific standards as reviewer.

A few statements in the *general replies* explain your a few comments in more detail.

**Statement 1**: A native English speaker check recommended. (Despite you suggest this, you made again a grammar mistake)

**Reply 1:** Papers English was checked by American native speaker earlier. But the English language would deserve an additional check.

**Statament 2:** Very bad structured manuscript body

**Reply 2:** I would recommend again to check publications written below:

Example 1: Avalanche hazards and mitigation in Austria: A review (Holler, 2007),
Example 2: Snow Avalanche Hazard in Canada – a Review
(https://doi.org/10.1023/A:1022998512227)

In terms of body structure body structure of "Assessment of avalanche hazard situation in Turkey during years 2010s" is similar with them. And there is gap on this issue in Turkey

**Statement 3:** Reader couldn't find any data from 2010s

**Reply 3** : Thanks to make this comment. Even, there are not many avalanche events. I will add avalanche events from 2017, 2018 if I get to change to revise paper.

*With my best regards,*
*Tayfun Kurt*

---

## Referee Comment (RC3) · Anonymous Referee #2 · 2 Oct 2018

Dear Author, I have read your response to my comments. Normally it is expecting from an author to adress point by point to the comments raised by the reviewer(s). Unfortunately, in your case you prefer to insult reviewer instead of try to improving the quality of your manuscript. As you know very well that the peer review process for a journal publication is essential, and works as a quality control mechanism. At the same time it is a process by which experts evaluate scholarly works based on their expertise and its aim is to ensure a high quality of published science. Please keep in mind that, as reviewers we don't decide to accept or reject papers but we recommend a decision. Final decision only belong to the journal editors. After this short informative introduction I would like to emphasize that peer review process is perfect chance for an author to improve the quality of his/her paper. As an author many times my papers rejected. After

revising based on reviewers' comments and editor's decision I submitted to another journal and find oppurtunity to publish it. In your case, in my opinion your paper not mature enough to go to publication. You avoid to response my comments point by point. There are many wrong information in your paper which I already adressed some of them in my previous report. I listed two more examples below: 1. Abstract (page 1, line 11): "……..., an avalanche occured in Torul, Köstere, in the province of Giresun, ……". Torul is a a town and district of Gümüşhane province not Giresun province. The distance between Gümüşhane and Giresun is about 170 km. Interestingly you gave this information only at abstract section. We couldn't see details at the main manuscript body. 2. Page 5 Table 1: Translation of the names of the two departments completely wrong. One of them is "Departments of Art Structures" ant the another is "Art of Project Management Structures Branch". I guess the reason of mistranslation is the word of "Sanat" in Turkish means in english "Art" and "YapÄślarÄś" is mean "strucures". This is true but for it is first meaning. If you use "Sanat YapÄślarÄś" terms in context of engineering (and particular for civil engineering) the meaning is "Hydraulic structures" not "Art structures". Actually "Art structures" meaningless in english when you try to mean culverts, bridges and other hydraulic structures.

Finally, ones more, i) THERE IS contradiction between title and content, ii) Unfortunately, manuscript DOES not elaborated well and iii) I have TO suggest to reject it.

Sincerely yours.

---

## Author Comment (AC3) · 3 Oct 2018

Dear Reviewer 2

Thanks to your valuable comments. I have already post my response earlier.

Best regards,

---

## Author Comment (AC4) · 3 Oct 2018

Dear Reviewer 2

Thanks to you.

Best regards,